# Analytical Bias in the Measurement of Plasma 25-Hydroxyvitamin D Concentrations in Infants

**DOI:** 10.3390/ijerph17020412

**Published:** 2020-01-08

**Authors:** Kristina Rueter, Lucinda J. Black, Anderson Jones, Max Bulsara, Michael W. Clarke, Cristina Gamez, Ee M. Lim, Debra J. Palmer, Susan L. Prescott, Aris Siafarikas

**Affiliations:** 1Division of Paediatrics, School of Medicine, The University of Western Australia, Perth 6009, Australia; anderson.jones@telethonkids.org.au (A.J.); cristina.gamez@telethonkids.org.au (C.G.); debbie.palmer@telethonkids.org.au (D.J.P.); susan.prescott@telethonkids.org.au (S.L.P.); aris.siafarikas@uwa.edu.au (A.S.); 2Perth Children’s Hospital, Department of Paediatric Immunology, Perth 6009, Australia; 3inVIVO Planetary Health, Group of the Worldwide Universities Network (WUN), West New York, NJ 07093, USA; 4School of Public Health, Curtin University, Perth 6102, Australia; lucinda.black@curtin.edu.au; 5Telethon Kids Institute, University of Western Australia, Perth 6009, Australia; 6Institute for Health Research, University of Notre Dame, Fremantle 6160, Australia; max.bulsara@nd.edu.au; 7Metabolomics Australia, Centre for Microscopy, Characterisation and Analysis, The University of Western Australia, Perth 6009, Australia; michael.clarke@uwa.edu.au; 8Sir Charles Gairdner Hospital, Department of Endocrinology, Perth 6009, Australia; EeMun.Lim@health.wa.gov.au; 9PathWest Laboratory Medicine, QEII Medical Centre, Nedlands 6009, Australia; 10Perth Children’s Hospital, Department of Paediatric Endocrinology and Diabetes, Perth 6009, Australia

**Keywords:** 25-hydroxyvitamin D, analytical bias, infants, vitamin D

## Abstract

Hypovitaminosis D is prevalent worldwide; however, analytical bias in the measurement of circulating 25-hydroxyvitamin D (25(OH)D) concentrations may affect clinical treatment decisions and research. We performed parallel plasma 25(OH)D analyses using the Abbott Architect i2000 chemiluminescent immunoassay (CIA) and liquid chromatography–tandem mass spectrometry (LC–MS/MS) for paired samples from the same infants at 3 (*n* = 69), 6 (*n* = 79) and 12 months (*n* = 73) of age. To test agreement, we used Lin’s concordance correlation coefficient and corresponding 95% confidence interval, Bland–Altman’s limits of agreement, and Bradley–Blackwood (BB) test. Agreement was high at 3 months (coefficient between difference and mean −0.076; BB F = 0.825; *p* = 0.440), good at 12 months (−0.25; BB F = 2.41; *p* = 0.097) but missing at 6 months of age (−0.39; BB F = 12.30; *p* < 0.001). Overall, 18 infants had disparate results based on the cut-off point for vitamin D deficiency (25(OH)D < 50 nmol/L), particularly at three months, with seven (10%) infants deficient according to CIA but not LC–MS/MS, and four (6%) deficient by LC–MS/MS but not CIA. To our knowledge, this is the first study to show that the reported 25(OH)D concentration may be influenced by both age and assay type. Physicians and researchers should be aware of these pitfalls when measuring circulating 25(OH)D concentrations in infants and when developing treatment plans based on measured vitamin D status.

## 1. Introduction

Vitamin D deficiency is a significant global concern due to both its high prevalence in diverse populations and its multisystem health implications [1,2,3,4,5]. Sunlight exposure is required for vitamin D synthesis, and a growing body of evidence suggests that lifestyle changes, including reduced outdoor activity, may explain the global rise of vitamin D insufficiency over the last decades [6,7,8]. There is long-standing awareness of the importance of vitamin D for bone health [4]. There is also expanding evidence that vitamin D plays a role in various other inflammatory non-communicable diseases [2,3,4].

Many of these multisystem effects may be influenced by the impact of vitamin D on the immune function [9]. Increased knowledge of the broader health implications of vitamin D status has translated to higher demands for measuring circulating 25-hydroxyvitamin D (25(OH)D) concentrations, with substantial cost to health systems [10,11]. This underscores the importance of cost-effective, reliable and accurate assays for determining vitamin D status—and a clear understanding of the variability between different assays used in clinical practice and research.

Thresholds for vitamin D deficiency are based on total 25(OH)D concentrations (25(OH)D_3_ plus 25(OH)D_2_) [12,13]. However, the accurate assessment of vitamin D status is challenging, since 25(OH)D is lipophilic, strongly bound to vitamin D-binding protein and has more than 50 epimers [12,14]. The interpretation of total circulating 25(OH)D concentrations is further complicated by the presence of the C3-epimeric form (C3-epi-25(OH)D_3_), particularly in infants aged ≤12 months [15,16]. Due to their molecular similarity, C3-epi-25(OH)D_3_ may be incorrectly interpreted as 25(OH)D by some assays. Furthermore, the downstream hydroxylated C3-1,25(OH)_2_D is thought to have considerably lower biological activity than the active form of vitamin D, 1,25-dihydroxyvitamin D. Inadvertently including C3-epi-25(OH)D_3_ may result in an overestimation of circulating 25(OH)D concentrations in infants, leading to under-prescription of supplements and undue toxicity concerns [15,16].

A liquid chromatography–tandem mass spectrometry (LC–MS/MS) method that is certified to the reference measurement procedures (RMPs) developed by the National Institute of Standards and Technology, Ghent University, and the US Centers for Disease Control and Prevention [17,18] is considered to be the gold standard for measuring circulating 25(OH)D concentrations [19]. LC–MS/MS can report 25(OH)D_3_ concentrations independently of 25(OH)D_2_ and C3-epi-25(OH)D_3_ [12]. However, the requirement of highly trained technologists, the limited sample throughput, and the high cost of the equipment, places constraints on the widespread use of LC–MS/MS in clinical laboratories. In contrast, the automated immunoassay is less expensive and much easier to perform with high-throughput capacity [20]. This is the method most commonly used in clinical laboratories and is the basis for the clinical classification of vitamin D deficiency and subsequent recommendations for vitamin D supplementation [12].

Despite the introduction of monitoring programs (International Vitamin D External Quality Assessment Scheme (DEQAS) [21], Vitamin D Standardization Program (VDSP) [17,22] and the College of Pathologists’ recommendations [23]), substantial between-assay differences have been documented between laboratories engaged in measuring circulating 25(OH)D concentrations [24]. Studies in adults have demonstrated that serum 25(OH)D concentrations measured by LC–MS/MS were generally higher than those measured by other assays [19,24].

Limited data on assay variability are available in infants [25,26]. An Australian study in neonates showed a tendency towards lower serum 25(OH)D concentrations measured by LC–MS/MS compared with those determined by immunoassay, although the difference was not statistically significant [25]. Adequate vitamin D status in infancy has unique implications for long-term health—not only for life-long bone health [27], but also for immune and metabolic health [28]. Even small individual differences around the cut-off point for vitamin D deficiency can potentially result in misclassification of vitamin D status [26] and alter decisions around vitamin D supplementation [29]. Given the lack of data comparing the performances of vitamin D assays in infants, we aimed to investigate the inter-assay variability between the chemiluminescent immunoassay (CIA) and the LC–MS/MS method in infants at 3, 6 and 12 months of age.

## 2. Materials and Methods

This is an observational study comparing parallel analyses by CIA and LC–MS/MS of identical paired samples for plasma 25(OH)D concentrations in the same cohort of infants at 3, 6, and 12 months of age. The samples were collected between March 2012 and August 2016 as part of a randomized controlled trial conducted in Perth, Australia [30], which was designed to assess the effect of oral vitamin D supplementation (400 IU/day cholecalciferol over 6 months) on infant immune outcomes (ACTRN12612000787886). Written informed consent was obtained before trial participation. Ethics approval was granted by the Human Research Ethics Committees of Princess Margaret Hospital (1959EP).

Peripheral blood was collected by venipuncture and processed for plasma and mononuclear cells. Ref. [30] Two different assays were used for the determination of plasma 25(OH)D concentrations.

We used a delayed one-step chemiluminescent microparticle immunoassay (CIA) on an automated Abbott Architect i2000 (Abbott Laboratories, Illinois). The calibrators were standardized against Standard Reference Material 2-972 from the National Institute of Standards and Technology [31]. The analytical coefficient of variation for 25(OH)D was 7.8% at 24 nmol/L, 4.2% at 46 nmol/L and 4.1% at 82 nmol/L. Cross reactivity between 25(OH) vitamin D and 25(OH)D_3_ epimer was 1.3%, and that between 25(OH) vitamin D and 25(OH)D_2_ epimer was 0.8%. The Abbott Architect i2000 is accredited by the National Association of Testing Authorities for measurement of 25(OH)D, and the recently improved version has been shown to have reduced variability in comparison to LC–MS/MS [32].

The concentrations of 25(OH)D were also analysed by the LC–MS/MS method certified by the VDSP [17]. This method has been previously described [33]. Briefly, liquid–liquid extraction of vitamin D metabolites was followed by 2-dimensional LC–MS/MS analysis on an Agilent 6460 LC–QQQ mass spectrometer. The coefficient of variation of the assay was consistently <5% at 23 to 182 nmol/L and could report down to 2 nmol/L. Using this assay, 25(OH)D_2_, 25(OH)D_3_ and C3-epimer concentrations were measured; 25(OH)D_2_ concentrations were consistently <3 nmol/L and were therefore not reported [33]. As Abbott Architect i2000 did not measure C3-epimer concentrations, these were not included in our analysis, however, they are reported in Table 1 and Table 2.

Statistical analysis was conducted using Stata v14 (ref: StataCorp. 2015. Stata Statistical Software: Release 14. College Station, TX: StataCorp LP). As measures of agreement, we used Lin’s concordance correlation coefficient and corresponding 95% confidence interval and Bland–Altman’s limits of agreement method. Bradley–Blackwood test, a test of equality of means and variances of both variables, was also performed. All statistical tests were two-tailed. Statistical significance was set at *p* < 0.05. Concordance was implied if the value for the correlation between difference and mean was near zero and the Bradley–Blackwood coefficient (F) was not statistically significant. Vitamin D deficiency was defined as 25(OH)D concentrations < 50 nmol/L, the preferred cut-off point used in Australia [34,35]. We compared results for plasma 25(OH)D concentrations between CIA and LC–MS/MS. For each assay, we also determined the proportion of samples identified as below the cut-off point for vitamin D deficiency (25(OH)D < 50 nmol/L) [36,37].

## 3. Results

Of the 120 participants, parallel plasma 25(OH)D measurements using CIA and LC–MS/MS were available for paired samples of 69 infants at 3 months, 79 infants at 6 months, and 73 infants at 12 months of age (Table 1).

The Abbott CIA reported the sum of D_2_ and D_3_ metabolites. Using LC–MS/MS, we were able to differentiate D_2_ and D_3_ metabolites, and this analysis consistently revealed 25(OH)D_2_ concentrations of <3 nmol/L for each included sample.

There was good agreement between assays at three months of age (Bland–Altman bias =2.63, correlation between difference and mean R = −0.076; Bradley–Blackwood F = 0.825; *p* = 0.443) (Figure 1) but not at six months of age (Bland–Altman bias= 6.316; R = −0.40; F = 12.32; *p* < 0.001) (Figure 2). There was agreement at 12 months of age (Bland–Altman bias= 0.433, R = −0.25; F = 2.41; *p* = 0.097) (Figure 3).

For each assay, we also determined the proportion of samples identified as below the cut-off point for vitamin D deficiency (25(OH)D < 50 nmol/L). Seven (10%) of the three-month-old infants were defined as vitamin D-deficient by CIA but not LC–MS/MS, and four (6%) were defined as deficient by LC–MS/MS but not CIA (Table 2). Among the 6- and 12-month-old infants, five and two infants, respectively, were deficient according to laboratory CIA but not LC–MS/MS. None of the 6- and 12-month-old infants were vitamin D-deficient according to LC–MS/MS. (Table 2).

## 4. Discussion

This study compared the two major 25(OH)D assays currently in use (immunoassay and LC–MS/MS) for analysing plasma 25(OH)D concentrations in the same cohort of 3-, 6-, and 12-month-old infants. We found that inter-assay variation differed by age. Overall, there was agreement at 3 and 12 months, but not at 6 months of age. On an individual basis, measurements around the cut-off point for vitamin D deficiency (25(OH)D < 50 nmol/L) differed between assays. At three months of age, 10% of the infants were defined as vitamin D-deficient by CIA but not LC–MS/MS, and 6% were defined as deficient by LC–MS/MS but not CIA. Hence, such assay variability may lead to therapeutic consequences for younger infants, including errors in the determination of the need and dose of vitamin D supplementation to promote bone health.

Uniquely, we compared CIA and LC–MS/MS longitudinally in the first year of life and noticed an age difference in assay variability which is supported by a European study in older children and adults [6], suggesting that this finding may be influenced by age-related metabolic changes. The observed age difference in assay variability in our cohort may also be impacted by the actual 25(OH)D concentration, as other studies have observed more agreement for lower than for higher concentrations of 25(OH)D [24,38]. In our study, 25(OH)D concentrations were higher at 6 months than at 3 and 12 months of age, which could be explained by nutritional factors, such as changing from breastfeeding to vitamin D-enriched formulas. Further, vitamin D supplementation commenced in some infants at three months of age due to low vitamin D status, which likely resulted in higher plasma 25(OH)D concentrations at six months.

Potentially, it could be argued that the C3-epimer concentrations may have affected the outcome of the comparison between the two assays. However, in our study, C3-epimer concentrations were very similar at three and six months of age, although our results showed agreement between methods at three months of age but not at six months of age. Furthermore, looking at CIA, cross reactivity between 25(OH)D_3_ and the C3-epimer is known to be only minimal (1.3%). Hence, detection of the C3-epimer would not explain the missing agreement between the two assays at six months of age. 

While our results are consistent with those of the previous small (*n* = 10) South Australian study on neonates [25], there are a number of points of distinction. Firstly, the South Australian study used an enzyme immunoassay for comparison with LC–MS/MS, while we compared the more commonly used CIA. Secondly, in the South Australian study, a capillary blood sample was collected for a newborn screening test, while we used venous blood samples from older infants. Thirdly, we analysed plasma 25(OH)D concentrations at three different time points through the first year of life, while the South Australian study only collected a single sample during the neonatal period. Hence, for the first time, we were able to track inter-assay differences in infancy longitudinally.

Using blood samples from children aged 9–11 years, an Iranian paediatric study compared a competitive protein-binding assay-based enzyme immunoassay (CPBA) with a high-pressure liquid chromatography (HPLC) method for measuring 25(OH)D concentrations [39]. They found that CPBA sensitivity and specificity were poor (44% and 61%, respectively) compared to HPLC. Although the results of this study cannot be directly compared to ours (different assays, ages, ethnicity and lower cut-off point for vitamin D deficiency at 12.5 nmol/L) [19,38], its findings nonetheless support that vitamin D determinations in infants are influenced by the assay used. A recent German study which includes children aged 1 to 17 years highlights how standardization has a substantial impact on estimates of vitamin D status. The authors concluded that standardization of clinical, research laboratory and commercial assays for 25(OH)D measurement is urgently required, supporting our findings that until then, researchers and clinicians need to be aware of the problems when using different assays [6].

Studies in adults also demonstrate inter-assay variability [17,19,24,38,40]. Due to differences in sample preparation and lack of standardization in calibration, there is also substantial variability between the same assay type in different laboratories [13,19,25,41,42]. Substantial variation was even observed using repeated measures of the same assay in the same laboratory [38].

Since accurate and reliable measures are essential for clinical indications for vitamin D supplementation, assay variability may have an influence on clinical decisions. Correct clinical decisions, while important at all ages, are particularly important in early childhood [43,44], a life stage when optimal bone health is crucial for gross motor development. Furthermore, there is growing evidence that vitamin D deficiency may have a long-lasting impact on immune and metabolic health. This highlights the need for reliable and standardized measurements and, if required, adequate vitamin D supplementation early in life [28]. Ideally, the same assay should be used longitudinally to monitor treatment responses.

Assay variability may also impact on research, since the use of different assays hinders pooling of results. A few randomized controlled trials have assessed the efficacy of daily drops of 400 IU vitamin D in the first year of life. However, different assays were used to quantify 25(OH)D: CIA [30,45], radioimmunoassay (RIA) [46] and LC–MS/MS [47], and only two of these studies referred to the use of an external quality assessment system [30,47].

Although efforts have been made to standardize 25(OH)D measurements in research and clinical settings [17,21,22,42], substantial variability in the measurement of 25(OH)D concentrations still occurs. In these instances, harmonising results to international standards by reanalysing a relatively small number of samples is suggested [24].

A strength of the present study is the analysis of sample pairs in the same cohort of infants at 3, 6 and 12 months of age. However, our study was a single-centre study with a limited number of analysed samples; therefore, results may not be generalizable. The inherent limitations of vitamin D assays must also be considered when comparing variability, including variations between and within methods carried out in different laboratories, by different operators, or using different reagent kit batches.

## 5. Conclusions

We have shown that age in infancy influences the variability between commonly used assays for measuring 25(OH)D concentrations. Physicians, researchers, laboratory technicians and authorities need to be aware of the current limitations when interpreting and comparing vitamin D values derived from different assays. Accurate measurement of circulating 25(OH)D concentrations is essential for indicating the need for vitamin D supplementation, which is particularly important in pregnancy, lactation and infancy, when optimal bone growth is of paramount importance. Reliable measurement is also crucial for research and public health initiatives, particularly for estimating the prevalence of vitamin D deficiency and for following participants longitudinally.

## Figures and Tables

**Figure 1 ijerph-17-00412-f001:**
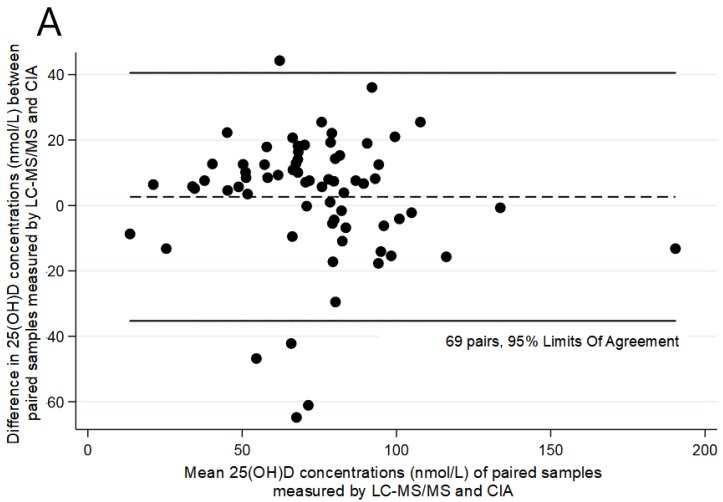
Comparison of 25(OH)D concentrations in paired samples at three months of age using two different assays: (**A**) Bland–Altman plot; (**B**) Concordance analysis; 25(OH)D, 25-hydroxyvitamin D; Lin’s correlation = 0.760 (0.659–0.860, *p* < 0.001).

**Figure 2 ijerph-17-00412-f002:**
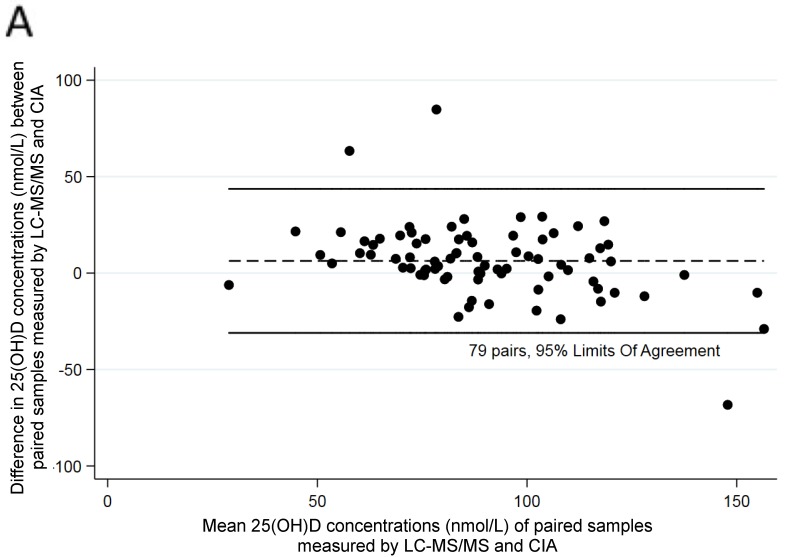
Comparison of assays in paired samples at six months of age: (**A**) Difference between two parallel analyses using Bland–Altman plots; (**B**) Concordance analysis; Lin’s correlation = 0.710 (0.610–0.810, *p* < 0.001).

**Figure 3 ijerph-17-00412-f003:**
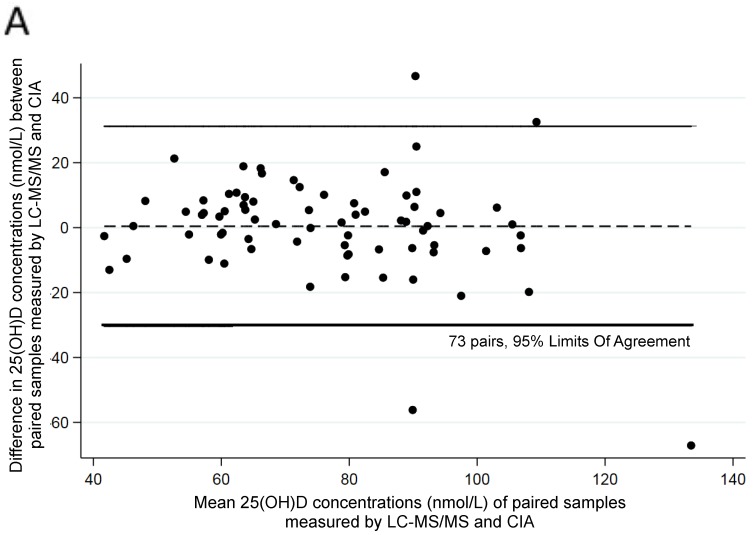
Comparison of assays in paired samples at 12 months of age: (**A**) Difference between two parallel analyses using Bland–Altman plots; (**B**) Concordance analysis; Lin’s correlation = 0.694 (0.578–0.811, *p* < 0.001).

**Table 1 ijerph-17-00412-t001:** Descriptive statistics for plasma 25(OH)D concentrations reported by chemiluminescent immunoassay (CIA) and liquid chromatography–tandem mass spectrometry (LC–MS/MS).

Assay	*n*	Range (nmol/L)	Mean (nmol/L)	SD (nmol/L)	Mean (SD) Difference between Laboratories
**Age: 3 months**					
**CIA**	69	18.00–97.00	72.16	28.81	
**LC–MS/MS** C3-epimer	69	9.30–183.80 2.40–33.00	74.79 11.11	27.43 7.40	+2.63 (19.35)
**Age: 6 months**					
**CIA**	79	26.00–182.00	86.91	29.29	
**LC–MS/MS** C3-epimer	79	25.80–149.80 2.60–36.00	93.23 10.92	22.18 7.32	+6.32 (19.04)
**Age: 12 months**					
**CIA**	73	42.00–167.00	76.19	21.83	
**LC–MS/MS** C3-epimer	73	36.00–125.60 1.80–16.20	76.62 4.53	18.19 2.56	+0.43 (15.71)

SD, standard deviation.

**Table 2 ijerph-17-00412-t002:** Individual plasma 25(OH)D concentrations of infants with inconsistent measurements at the cut-off point for vitamin D deficiency comparing CIA and LC–MS/MS data.

Age	CIA (nmol/L)	LC–MS/MS (nmol/L)
3 months	40 *	84.3 (9 ^∧^)
	46 *	56.2 (3.6 ^∧^)
	44 *	56.6 (6.9 ^∧^)
	49 *	66.9 (15 ^∧^)
	49 *	53.5 (2.6 ^∧^)
	34 *	56.3 (5.3 ^∧^)
	46 *	51.7 (4.3 ^∧^)
	78	31.2 * (4.0 ^∧^)
	102	40.9 * (3.4 ^∧^)
	89	35.2 * (3.6 ^∧^)
	87	44.8 * (2.4 ^∧^)
6 months	36 *	120.8 (2.7 ^∧^)
	34 *	55.6 (4.4 ^∧^)
	46 *	55.4 (3.8 ^∧^)
	45 *	66.2 (4.8 ^∧^)
	26 *	89.3 (10.6 ^∧^)
12 months	44 *	52.2 (2.4 ^∧^)
	42 *	63.3 (3.0 ^∧^)

* Below the cut-off point for vitamin D deficiency (25(OH)D <50 nmol/L); ^∧^ including C3-epimeric form.

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
