# Peer review of "Analytical Bias in the Measurement of Plasma 25-Hydroxyvitamin D Concentrations in Infants"

_ijerph, 2020, doi:10.3390/ijerph17020412_

Round 1
Reviewer 1 Report
In the abstract: I think it better if you write that you compare LC-MS/MS with the Abbott Architect i2000 chemiluminescent immunoassay.
General questions:
1. I guess it wasn´t the same children/individuals who were examined at 3, 6 and 12 months or was it.? This must be clarified booth in abstract and in the method.
2. In the introduction and the method part of the paper you write about the C3-epimeric forms and this is of course very relevant. You also write that you measured the concentration of the C3 epimer with LC-MS/MS, but you do not report this concentration in the result section. It would have fit in well for example in Table 2 to better understand the differences in 25(OH)D concentrations found between the LC-MS/MS and the CIA assays.
3. Methods:The infants participated in a randomized controlled trial. What was the dose oral vitamin D they got? Cholecalciferol or ergokalciferol? Please clarify
4. Methods. Samples were taken between march 2012 and August 2016. Could differences between assays and 25(OH)D concentration depend on seasonal variation in sampling?
5. Discussion line 163.You write that inter-assay variation differed by age. But it was not the same children you followed I guess? The difference found at children aged 6 months could depend on various reasons in these children, eg high concentrations of C3-epimer, disease etc?
6. Discussion line 196. You write that 25(OH)D concentrations are generally lower when measured by immunoassays than those measured by LC-MS(/MS. I am not sure this is correct. The DEQAS reports for this year show only small differences between immunoassays and LC-MS/MS.
7. In summary, I think it would be valuable if you could better describe how the study really was done. When was the samples taken? Was it different kids at 3, 6 and 12 months? If they were given vitamin D-supplementation etc. Could you also speculate on the differences between LC-MS/MS and the Abbott immunoassay. It would be valuable if you could report the C3 epi 25(OH)D concentrations.
Reviewer 2 Report
I would like to thank the authors for the effort in research design and to address the methodology proposed in this research. The comments provided could improve the present manuscript and the interest of the readers.

Round 2
Reviewer 1 Report
The paper is know significantly improved. No futher comments.
Author Response
Thank you kindly for your earlier suggestions and for confirming that the changes made were satisfactory.